# Anti-Inflammatory Activity of Gomphrenin-Rich Fraction from *Basella alba* L. f. *rubra* Fruits

**DOI:** 10.3390/nu16244393

**Published:** 2024-12-20

**Authors:** Agnieszka Rusak, Monika Mrozowska, Weronika Kozłowska, Benita Wiatrak, Piotr Dzięgiel, Sławomir Wybraniec, Ivana Carev, Agnieszka Jama-Kmiecik, Adam Matkowski, Sylwia Zielińska

**Affiliations:** 1Division of Histology and Embryology, Department of Human Morphology and Embryology, Faculty of Medicine, Wroclaw Medical University, Chałubińskiego 6a, 50-368 Wroclaw, Poland; agnieszka.rusak@umw.edu.pl (A.R.); monika.mrozowska@umw.edu.pl (M.M.); piotr.dziegiel@umw.edu.pl (P.D.); 2Division of Pharmaceutical Biotechnology, Department of Pharmaceutical Biology and Biotechnology, Faculty of Pharmacy, Wroclaw Medical University, Borowska 211, 50-556 Wroclaw, Poland; weronika.kozlowska@umw.edu.pl; 3Department of Pharmacology, Faculty of Medicine, Wroclaw Medical University, J. Mikulicza-Radeckiego 2, 50-345 Wroclaw, Poland; benita.wiatrak@umw.edu.pl; 4Department of Chemical Technology and Environmental Analytics (C-1), Cracow University of Technology, Warszawska 24, 31-155 Cracow, Poland; slawomir.wybraniec@pk.edu.pl; 5Faculty of Chemistry and Technology, University of Split, Ruđera Boškovića 35, 21 000 Split, Croatia; ivana.carev@ktf-split.hr; 6Department of Medical Biology, Faculty of Nursing and Midwifery, Wroclaw Medical University, Chałubińskiego 3, 50-368 Wroclaw, Poland; agnieszka.jama-kmiecik@umw.edu.pl; 7Division of Pharmaceutical Biology and Botany, Department of Pharmaceutical Biology and Biotechnology, Faculty of Pharmacy, Wroclaw Medical University, Borowska 211A, 50-556 Wroclaw, Poland; 8Botanical Garden of Medicinal Plants, Department of Pharmaceutical Biology and Biotechnology, Faculty of Pharmacy, Wroclaw Medical University, Al. Jana Kochanowskiego 10/12/14, 51-601 Wroclaw, Poland

**Keywords:** *Basella*, betalain, gomphrenin, anti-inflammatory, antioxidant

## Abstract

**Background/Objectives:** *Basella alba* L. (Malabar spinach, Basellaceae), widely consumed as a leafy vegetable, produces dark-colored fruits rich in betacyanins, including rare 6-glycosylated derivatives called gomphrenins. Comprehensive studies on the anti-inflammatory potential of its gomphrenin fraction (A) and crude extract (B) employed various analytical and biological methods. **Methods:** Cytotoxicity and anti-inflammatory effects were tested on human and animal cell models using SRB, DCF-DA, Griess, MDA, and ELISA assays. **Results:** Both the crude extract and enriched gomphrenin fraction exhibited significant anti-inflammatory and antioxidant effects in vitro. They inhibited pro-inflammatory cytokines IL-1β and IL-6, reduced oxidative stress markers (ROS, NO), and decreased lipid peroxidation. The enriched gomphrenin fraction (A) showed stronger antioxidant and anti-inflammatory effects, particularly in reducing ROS and NO levels, though not always concentration-dependent. Both A and B inhibited NF-κB and COX activity. **Conclusions:** These findings highlight the therapeutic potential of *B. alba* f. *rubra* fruit extract and betalain fraction for inflammation-related conditions, encouraging further exploration of their mechanisms and application.

## 1. Introduction

Betalains, a class of water-soluble pigments, are found in plants of the Caryophyllales order, including the families Basellaceae, Cactaceae, Nyctaginaceae, and Amaranthaceae, as well as in volva-type mushrooms such as *Amanita muscaria* (L.) Lam. [1,2,3,4]. These compounds are divided into two classes, betacyanins (red-violet pigments, examples of structures are shown in Figure 1) and betaxanthins (yellow-orange pigments). 

Gomphrenins, a specific type of betacyanin, were first identified in *Gomphrena globosa* L. (globe amaranth), a plant species of the Amaranthaceae family [5]. Betalains, including gomphrenins, have garnered scientific interest for their potential anti-inflammatory properties, pertinent to numerous chronic diseases characterized by inflammation. The anti-inflammatory effects of these compounds are attributed to several mechanisms such as inhibition of pro-inflammatory enzymes, antioxidant activity, modulation of the NF-κB pathway, and suppression of inflammatory cytokines [6,7,8]. Betalains inhibit pro-inflammatory enzymes, such as cyclooxygenase-2 (COX-2), which is responsible for the synthesis of prostaglandins, which promote inflammation, pain, and fever. For instance, betanin, a major betacyanin from beets (*Beta vulgaris* L.), has been shown to reduce COX-2 activity, thereby diminishing the production of these pro-inflammatory mediators [9]. The significant antioxidant properties of betalains are also important because they participate in the neutralization of reactive oxygen species (ROS) and a reduction in oxidative stress, which is closely linked to inflammation. By scavenging free radicals, gomphrenins and other betalains can mitigate the oxidative damage that exacerbates inflammatory responses [10]. Another aspect is the possibility of influencing the modulation of the NF-κB pathway, which is a key regulator of inflammation. Betalains inhibit the activation of NF-κB, leading to a decrease in the secretion of various inflammatory cytokines, such as tumor necrosis factor-alpha (TNF-α) and interleukin-6 (IL-6) [11]. Furthermore, betalains downregulate the production of pro-inflammatory cytokines. For example, betalains from *Basella alba* L. (Basellaceae), *Gomphrena globosa* L. (Amaranthaceae), *Mirabilis jalapa* L. (Nyctaginaceae), and *Opuntia ficus-indica* (L.) Mill. (Cactaceae) were previously found to reduce cytokine levels, mitigating inflammation in cellular models [6,11]. Interestingly, under certain conditions, betalains may exhibit pro-inflammatory effects [6]. This pro-oxidant behavior can lead to increased ROS production, potentially triggering inflammatory pathways. This dual role has been noted in studies where high doses of betalains induced oxidative stress. The effects of betalains can vary depending on the cellular context and concentration. While moderate concentrations typically exert anti-inflammatory effects, higher concentrations might activate inflammatory pathways or disrupt cellular homeostasis, leading to pro-inflammatory outcomes. This phenomenon underscores the importance of dosage in the therapeutic and preventive application of betalains [12].

The existing studies predominantly focus on general betalain types or specific sources, such as beet-derived betanin, with limited exploration of less common betalain-rich sources. Thus, in this study, we have investigated the anti-inflammatory properties of a phytochemically characterized crude extract and enriched fraction of gomphrenins obtained from *B. alba* L. fruits.

As a source of betacyanin-rich material, we used the morphological form with burgundy-colored stems and very dark fruits. This form has no firmly established systematic position and is called *B. alba* f. *rubra*, considered a variety (*B. alba* var. *rubra),* or even considered a separate species (*Basella rubra* L). However, the current accepted nomenclature lists only the species *B. alba* L., and all other binomial names are not acknowledged.

The cytotoxicity and anti-inflammatory effects of the extracts and gomphrenins were assessed using normal human dermal fibroblasts (NHDFs), mouse fibroblasts (L929), and differentiated human acute leukemia monocyte (THP-1) cells. Various assays, including SRB, DCF-DA, Griess, MDA, and ELISA, were employed to evaluate cell viability, oxidative stress markers, and cytokine levels, providing detailed insights into the bioactivity of gomphrenins. A dual role of betalains, where higher concentrations may induce pro-inflammatory effects, remains underexplored, especially in the context of therapeutic applications where dose optimization is crucial.

## 2. Materials and Methods

### 2.1. Plant Material, Substances, and Reagents Used in Assays

*B. alba* f. *rubra* fruits were obtained from the greenhouse of the Agricultural University of Krakow (Faculty of Biotechnology and Horticulture) where sowing of the seeds was performed in a 3:1 ratio of soil and coconut pith mass and watered daily. The seedlings were transplanted to fertile soil with plenty of organic matter and a pH of 6.5–6.8. The plants were designed to support the fast climbing of the vines which were trellised so that they reached up to 3 m in length at controlled humidity and temperature, ensuring continuous flowering and fruiting [1].

The fruits from which extracts and enriched fractions were prepared were collected and prepared for phytochemical analysis by squeezing, centrifuging, and filtering through a 0.2 mm filter. The filtrate was then diluted three times with water and immediately analyzed for pigments or stored at −20 °C for later use. The extract was purified by open column chromatography on strongly acidic cation exchange resin (Strata X-C, Phenomenex, Torrance, CA, USA) as described previously [1]. The betacyanin fraction eluted with water was concentrated using a rotary evaporator under reduced pressure at 25 °C and purified by flash chromatography on silica C18 sorbent (Chromabond, Macherey-Nagel, Düren, Germany) as described previously [1], except that the gomphrenin fraction was eluted using an eluent composed of water/acetone/formic acid, 97/2/1 (*v*/*v*/*v*). The pigment fraction was then concentrated using a rotary evaporator under reduced pressure, resulting in gomphrenin enriched fraction designated with the symbol A. The unpurified crude extract was assigned with the symbol B.

The *Basella alba* L. f. *rubra* samples were submitted to spectrophotometric as well as LC-MS analyses [1]. For the measurement of the total concentration of the pigments, the extracts were analyzed spectrophotometrically. The total concentration was expressed as mg gomphrenin equivalents/100 g of fresh fruits. Quantification of betacyanins was evaluated taking a molar extinction coefficient of ε = 65,000 M^−1^ cm^−1^ at 539 nm for betanin in spectrophotometric calculations [13]. The betacyanin profiles were established by LC-MS analyses according to a previous report [1].

### 2.2. Cell Lines

Normal human dermal fibroblasts (NHDFs) (Lonza, Basel, Switzerland) and mouse fibroblasts (L929) (ATCC, Old Town Manassas, VA, USA) were used to determine the cytotoxicity of the tested compounds and plant extracts. The human acute leukemia monocyte cell line (THP-1, ATCC) was used as an inflammation model [14]. The NHDF and L929 cells were grown in Dulbecco Modified Eagle Medium (DMEM) (Capricorn, Ebsdorfergrund, Germany); THP-1 cells were cultured in RPMI 1640 medium with B-mercaptoethanol (Sigma-Aldrich, St. Louis, MO, USA). All media were supplemented with 10% fetal bovine serum (FBS, Sigma-Aldrich), and 1% of L-glutamine with streptomycin and penicillin solution (Sigma-Aldrich). Cells were grown in a CO_2_ incubator under 5% CO_2_ and 95% humidity at 37 °C. The morphological cells and confluence were evaluated twice a week under a microscope. Differentiation of THP-1 cells into macrophages was conducted in complete RPMI-1640 medium with 100 nM of phorbol 12-myristate-13-acetate (PMA) (Sigma-Aldrich) [15,16].

### 2.3. Tested Compounds and Reference Substances

Stock solutions (20 mg/mL) were prepared in distilled H_2_O and stored at 4 °C until use, but not for longer than 6 months. An initial concentration of 40 µg/mL was prepared in a culture medium appropriate to the cell line. Immediately before the test, concentrations [0.001–40 µg/mL] of the tested compounds and extracts were prepared in appropriate media. The concentration range (0.0001–40 μg/mL) for the gomphrenin enriched fraction and crude extract was determined based on literature data and the need to encompass biologically relevant and sub-toxic concentrations.

A cell culture in a culture medium (blank test, KK) as well as culture fluid without the presence of cells (DMSO, KR) were included to assess baseline activity and ensure the absence of non-specific effects caused by the experimental setup. Positive controls included lipopolysaccharide (LPS), used to induce inflammation, providing a model for testing the anti-inflammatory properties of the studied substances. Meloxicam (Mel) and indomethacin (Ind) served as reference anti-inflammatory drugs, allowing for a comparison of the efficacy of the tested extracts to clinically established treatments. Ascorbic acid (Asc) and acetylsalicylic acid (aspirin, Asp) were incorporated as antioxidant standards to evaluate the antioxidative capacity of the extracts under conditions of oxidative stress. Plain culture medium (M) was used to account for any potential effects of the medium itself on the measured parameters.

The selection of these controls was aimed at providing a comprehensive framework for evaluating the specific effects of the tested compounds. LPS-induced inflammation models and oxidative stress conditions ensured that both anti-inflammatory and antioxidative activities could be robustly assessed. The inclusion of established pharmacological agents (Mel, Ind, Asc, Asp) further allowed for benchmarking the bioactivity of the extracts, ensuring comparability with widely recognized therapeutic standards.

### 2.4. Experimental Design

Prior to exploring the potential for other bioactivities, a wide range of concentrations (0.0001–40 μg/mL) of the gomphrenin fraction and crude extract were used to investigate their effects on cell viability in normal human (NHDF) and murine (L929) models used to evaluate the cytotoxic effects of the tested compounds under conditions mimicking normal physiological environments using the SRB assay. The goal was to identify concentrations that maintained cell viability above 80%, ensuring that the selected range would not induce significant cytotoxicity. The lower end of the range (0.0001–0.01 μg/mL) was chosen to explore potential bioactivity at trace levels, as some plant-derived compounds are known to exhibit significant effects even at very low concentrations. The higher concentrations (up to 40 μg/mL) allowed for assessing the maximum efficacy and potential dose–response relationships, while also accounting for the higher tolerance observed in robust cell models. Concentration ranges used in similar studies on betalain-rich extracts and their bioactive properties were reviewed. Comparable ranges were reported in studies evaluating anti-inflammatory and antioxidant effects, which informed the upper and lower limits of the selected range. By integrating these factors, the chosen concentration range was intended to provide comprehensive insights into the bioactivity of the gomphrenin fraction and crude extract across different biological contexts.

Following this, based on SRB assay results, the gomphrenin fraction and crude extract in the concentration range of 0.1–10 ug/mL were tested in an inflammation model system. To this aim, differentiated THP-1 cells were treated with test compounds for 24 h and then the supernatant was removed, the cell culture was washed, and 100 µg/mL lipopolysaccharide (LPS) was administered. After this time, biological tests were performed. The SRB assay was performed analogously as described in Section 2.5. In addition, the effect of the tested compounds on the level of free oxygen radicals (ROS; DCF-DA assay), nitric oxide (NO, Griess reagent), and lipid peroxidation (MDA assay) was assessed. Using ready-made ELISA kits, the level of cytokines (IL-1*β*, IL-6), NF-κB and cyclooxygenases (COX) was assessed as well.

### 2.5. SRB Assay

The cell proliferation was determined using SRB dye, which binds to proteins and informs about the amount of cellular protein. After 24 h, cells were seeded on 96-well plates (TPP); cells were fixed with a cold trichloroacetic acid (TCA) solution (Sigma-Aldrich) to a final concentration of 10% *w*/*v* for 1 h at 4 °C. The crude extract and gomphrenin fraction in the concentration rage of 0.0001–40 μg/mL were administered to the remaining culture plates for 24 h, 48 h, and 72 h at 5% CO_2_ and 95% humidity at 37 °C, after which the plates were similarly fixed. All plates were washed five times with running water and dried in air at room temperature (RT). Afterwards, 0.4% SRB solution in 1% *v*/*v* acetic acid (Sigma-Aldrich) was added for 30 min, which was then rinsed five times with a 1% acetic acid solution. The plates were air-dried at RT, and then the SRB dye connected to intracellular proteins was dissolved with 10 mM Trizma base for 30 min with stirring on a shaker. Finally, the absorbance was measured at 540 nm with a Varioskan LUX microplate reader (Thermo Fisher Scientific, Waltham, MA, USA) [17,18].

### 2.6. Protective Properties Against ROS or NO Formation

After incubation, cells with crude extract, gomphrenin fraction, tested compounds, and LPS were exposed for 1 h to oxidative stress induced by 100 μM H_2_O_2_ to evaluate the protective properties of gomphrenin against ROS, and with 100 μM SIN-1 (peroxynitrite generator) to evaluate the protective properties of gomphrenin against NO in 5% CO_2_, 95% humidity, and 37 °C. Finally, the DCF-DA, and Griess tests, for cells exposed to oxidative stress, were performed. The experimental setup ensured that the protective effects of the extracts and fractions against ROS and NO generation could be precisely evaluated. The pre-treatment step allowed for an assessment of the ability of the test compounds to prevent or mitigate oxidative and nitrosative damage, while the controls provided benchmarks for comparison.

### 2.7. Level of Reactive Oxygen Species and Nitric Oxide (NO)

The THP-1 cells were seeded on the multi-well plate at 1 × 10^6^ cells/well density in macrophage differentiation medium. After 24 h, freshly prepared testing compounds were added for 1 h, and H_2_O_2_ (100 µM) was added into testing wells and as a positive control. After 1 h of incubating, the medium was moved to new plates. Then, 25 µM of DCF-DA solution in MEM without serum and phenol red was added for 1 h in a CO_2_ incubator at 37 °C [19]. The ROS level was then assessed fluorometrically with excitation at 485 nm and emission at 535 nm using a Varioskan LUX microplate reader. Finally, a mixture of reagent A (1% sulfanilamide in 5% phosphoric acid) and reagent B (0.1% N-(1-naphthyl)-ethylenediamine dihydrochloride) was added to the previously transferred supernatant in a 1:1 volume ratio. The plates with solution were left for 20 min in the dark at RT, and the nitric oxide level was measured calorimetrically at 548 nm using a Varioskan LUX microplate reader.

### 2.8. DCF-DA Assay

The fluorescent dye, 2′,7′-dichlorofluorescein diacetate (DCF-DA), used to measure free radicals levels, was prepared fresh, in the form of a solution, before use, by dissolving 1 mg of DCF-DA in 2.05 mL of 100% ethanol and diluting it in deionized water to a final concentration of 10 μM. Solutions were made fresh before using the MEM medium without serum and phenol red. The ROS level was measured after a further 1 h incubation with DCF-DA solution using a Varioskan LUX microplate reader (λex. = 485 nm, λem. = 535 nm).

### 2.9. Griess Assay

The Griess assay was carried out to detect the presence of nitrite ions in the solution. Two reagents, 0.1% N-(1-naphthyl)ethylenediamine dihydrochloride and 1% sulfanilic acid, were combined at the same volume and mixed immediately before use. Then, 150 μL of the solution was transferred to a new plate, and 20 μL of a mixture of Griess reagents and 130 μL of deionized water was added for 30 min at RT. Nitrite level was measured with the Varioskan LUX at a wavelength of 548 nm.

### 2.10. MDA (Lipid Peroxidation Assay)

Cells were detached using a scraper and collected into prepared centrifuged tubes, where 5 mL of 5% TCA was added, and they were homogenized using an ultrasonic homogenizer. Then, the samples were centrifuged, and the supernatant was collected for further research. Thiobarbituric acid (TBA) was dissolved in 7.5 mL of glacial acetic acid to make 25 mL of TBA solution. Following this, 0.1 M MDA was prepared, diluting 10 μL of 4.17 M MDA in 407 μL distilled water, and 2 mM MDA standard was prepared and used for the series of dilutions, to create a standard curve of MDAs. The tested supernatants and MDAs were incubated in a water bath at 95 °C for 30 min with TBA solution. To stop the reaction, the samples were left on ice for 10 min and centrifuged at 4000 rpm for 5 min. The supernatant was added to a 96-well plate, and the absorbance at 530 nm was measured using a Varioskan Lux microplate reader.

### 2.11. Cytokine Levels

Cytokine levels were measured in the supernatant according to the instructions of the manufacturer of the Elisa kits. The levels of IL-1*β* (Abcam, Cambridge, UK) and IL-6 (Abcam) were measured according to the manufacturer’s instructions.

### 2.12. NF-κB Levels

To assess the level of NF-κB, a ready-made kit (Abcam) was used, according to which cell lysates were prepared and measurements were made at 450 nm using a Varioskan LUX microplate reader.

### 2.13. COX Activity

The cell pellet was homogenized in cold buffer (0.1 M Tris-HCl, pH 7.8, containing 1 mM EDTA) and centrifuged at 10,000× *g* for 15 min at 4 °C. The collected supernatant was placed on ice and directly used for the assay according to the instructions of the manufacturer of the ready-made COX activity kit (Cayman, Ann Arbor, MI, USA). COX1 and COX2 were calculated based on total COX activity. The absorbance was measured at 590 nm using a Varioskan LUX microplate reader.

### 2.14. Statistical Analysis

All experiments were performed in four independent biological replicates, each including four technical replicates to ensure statistical reliability. For the ELISA assays, material was prepared analogously from four independent replicates, with each plate containing duplicate or triplicate measurements for each sample, performed in accordance with the kit manufacturer’s instructions.

Normality of the distribution of variables was analyzed using the Shapiro–Wilk test. Statistical analysis was performed using ANOVA with Tukey’s post hoc test for SRB assay and DCF-DA results, while Kruskal–Wallis with post hoc Dunn’s test was used to analyze the Griess and MDA assays’ results and cytokine levels. For all tests, *p* > 0.05 was considered significant. Statistical analysis was performed using Statistica version 13.3 (Tibco, Palo Alto, CA, USA). Additionally, the effect sizes were calculated to show the strength of the relationship between the tested compounds and control/reference compound. The confidence intervals (CIs) were calculated and the results are given in Appendix A.

## 3. Results and Discussion

To achieve a comprehensive understanding of the anti-inflammatory potential of gomphrenins from *B. alba* f. *rubra*, we employed a variety of analytical and biological methods. The pigment identification in the crude and purified extract was previously accomplished using a High-Resolution Mass Spectrometric System (LC-Q-Orbitrap-MS) coupled with HPLC separation [1]. Three novel pigments, two hexosyl-hexosyl-betanidins and hexosyl-betanidin, and their isoforms as well as acylated betacyanins such as 6′-*O*-*E*-caffeoyl-gomphrenin (malabarin), 6′-*O*-*E*-sinapoyl-gomphrenin (gandolin), 6′-*O*-*E*-4-coumaroyl-gomphrenin (globosin), 6′-*O*-*E*-feruloyl-gomphrenin (basellin) and their isoforms were detected in *B. alba* f. *rubra* fruits [1]. Moreover, malonylated betanidin 6-*O*-*β*-glucosides and their acyl migration isomers as well as novel betacyanins acylated with nitrogenous substituents (C_9_H_8_NO_4_-gomphrenin, C_10_H_10_NO_5_-gomphrenin, C_8_H_6_NO_3_-gomphrenin, and C_7_H_8_NO_2_-gomphrenin) were also detected previously in the analyzed plant material [1]. Therefore, a unique plant material rich in various betacyanin derivatives was used in our study to investigate its antioxidant and anti-inflammatory potential.

Due to discovery of a series of novel betacyanins not reported in any plant before [1], the *B. alba* f. *rubra* fruit extract and gomphrenin enriched fraction were tested for their potential antioxidant and anti-inflammatory activity. Acylation with the nitrogenous acyl substituents was considered as a new phenomenon not only observed in the betacyanin group of compounds but also in polyphenolics. This fact, as well as the relatively high concentrations of acylated gomphrenins accompanied by polyphenolic compounds found in *B. alba* f. *rubra* fruits, makes this plant material a unique source of bioactive constituents that is worthy of investigation for future nutrient applications and medical purposes.

The cytotoxicity and anti-inflammatory effects of the gomphrenin enriched fraction and crude extract were tested using human and animal cell models. We utilized normal human dermal fibroblasts (NHDFs), mouse fibroblasts (L929), and differentiated human acute leukemia monocyte (THP-1) cells. A variety of assays, including SRB, DCF-DA, Griess, MDA, and ELISA, were used to measure cell viability, oxidative stress markers, and cytokine levels, offering comprehensive insights into the bioactivity of the gomphrenin enriched fraction and crude extract. Through these methods, we aimed to assess cell viability, oxidative stress, and cytokine production, thereby elucidating the bioactivity and therapeutic potential of the gomphrenin enriched fraction and crude extract.

This study demonstrated that both the crude extract and enriched fraction of gomphrenins from *B. alba* f. *rubra* exhibited significant anti-inflammatory properties in vitro in terms of IL-1β secretion. In turn, they did not show significant secretion inhibition of IL-6, reduced oxidative stress markers (ROS, NO), or decreased lipid peroxidation levels.

In the SRB assay for the L929 mouse fibroblast cell line, higher cell viability was observed with the crude extract compared to the gomphrenin enriched fraction after 24 h and 48 h of treatment. However, after 72 h, this trend reversed, with a larger number of live cells observed under the gomphrenin enriched fraction treatment. Despite these changes, no statistically significant difference was found between the crude extract and the purified gomphrenin fraction across all time points. Additionally, no dose-dependent effects were observed (Figure 2). For normal human dermal fibroblasts (NHDFs), significantly higher cell survival was consistently noted for the gomphrenin enriched fraction compared to the crude extract, irrespective of the incubation duration (Figure 3).

The results from the DCF-DA assay provide valuable insights into the antioxidative effects of the crude extracts and gomphrenin enriched fractions from *B. alba* f. *rubra* on differentiated THP-1 cells. The DCF-DA (2′,7′-dichlorofluorescein diacetate) assay was employed to evaluate the intracellular ROS levels in differentiated THP-1 cells treated with gomphrenin enriched fractions and crude extracts from *B. alba* f. *rubra*. The DCF-DA assay results, depicted in the provided bar graph, illustrate the levels of reactive oxygen species (ROS) in differentiated THP-1 cells under various treatment conditions.

We evaluated the levels of reactive oxygen species (ROS) using the DCF-DA assay under various treatment conditions (Figure 4a). The positive control with hydrogen peroxide (H_2_O_2_) showed a marked increase in ROS levels, confirming the expected induction of oxidative stress. Similarly, lipopolysaccharide (LPS) treatments, used as the control condition, for both 1 h and 24 h treatments also resulted in elevated ROS levels, demonstrating the oxidative stress associated with inflammatory stimuli.

When we assessed the crude extracts, at concentrations of 0.1, 1, and 10 μg/mL, we observed moderate ROS levels, which were generally lower than those seen in the LPS and H_2_O_2_ controls. The results indicated that both the gomphrenin enriched fraction and the crude extract significantly reduced ROS levels compared to the control treated with LPS alone. This suggests that the extracts and gomphrenin possess strong antioxidant properties, capable of scavenging free radicals and mitigating oxidative stress.

Specifically, cells treated with the crude extract exhibited a more pronounced decrease in ROS levels than those treated with the gomphrenin enriched fraction, indicating a higher efficacy of the purified fraction in reducing oxidative stress. This was consistent across different concentrations used in the assay, demonstrating the potent antioxidant capacity of gomphrenins alone.

Interestingly, while the gomphrenin enriched fraction did not show a clear concentration-dependent trend in reducing ROS, the crude extract exhibited a more consistent and pronounced reduction in ROS levels across all concentrations tested.

The DCF-DA assay results affirm the antioxidative potential of the extract and gomphrenin enriched fraction from *B. alba* f. *rubra*, which demonstrated a significant but moderate ability to reduce ROS levels.

In our study, we assessed lipid peroxidation using the MDA assay to evaluate the extent of oxidative damage in differentiated THP-1 cells treated with a crude extract and gomphrenin enriched fraction from *B. alba* f. *rubra*.

The positive control with hydrogen peroxide (H_2_O_2_) exhibited significantly elevated MDA levels, confirming the induction of substantial oxidative stress and lipid peroxidation. Similarly, lipopolysaccharide (LPS) treatments for both 1 h and 24 h showed increased MDA levels, indicating oxidative damage associated with inflammation (Figure 4b).

When examining the crude extract at concentrations of 0.1, 1, and 10 μg/mL, we noted that MDA levels were elevated but generally lower than those observed with H_2_O_2_ and LPS treatments. This suggests that the crude extract presents some protective effects against lipid peroxidation.

The crude extract, however, demonstrated even lower MDA levels compared to the gomphrenin enriched fraction across all concentrations tested. This indicated a more potent ability to prevent lipid peroxidation, highlighting the stronger antioxidative properties of the crude extract.

Both plant matrices exhibited a clear concentration-dependent trend in reducing MDA levels, at all tested concentrations. This consistency further emphasized the enhanced efficacy of the tested samples in preventing oxidative damage to lipids.

The data suggest that the crude extract has a superior capability to inhibit lipid peroxidation compared to the gomphrenin enriched fraction.

The Griess test results provided insights into the nitrite levels in differentiated THP-1 cells treated with various concentrations of the crude extract and gomphrenin enriched fraction of *B. alba* f. *rubra*, indicating the degree of nitric oxide (NO) production, which is a marker of inflammatory response. In our study, we employed the Griess test to measure nitrite levels as an indicator of nitric oxide production in differentiated THP-1 cells. The Griess test results revealed significant differences in nitrite levels across the different samples, reflecting the anti-inflammatory properties of the tested substances. The positive controls, sodium nitroprusside (SIN-1) and lipopolysaccharide (LPS) treatments, exhibited markedly elevated nitrite levels, indicating high NO production and confirming their role in inducing a strong inflammatory response. As expected, LPS treatment, both for 1 h and 24 h, resulted in substantial nitrite accumulation, demonstrating its potent pro-inflammatory effect. In contrast, treatments with the crude extract and gomphrenin enriched fraction at concentrations of 0.1, 1, and 10 µg/mL showed reduced nitrite levels compared to the positive controls. Both plant matrices showed a similar potential of NO level secretion inhibition to meloxicam and indomethacin, but lower than ascorbic acid (Figure 4c).

As mentioned before, the gomphrenin enriched fraction demonstrated even more pronounced reductions in nitrite levels across all tested concentrations. The results indicated that the crude extract of *B. alba* f. *rubra* did not exhibit a dose-dependent reduction in nitrite levels (Figure 4c). When comparing the nitrite levels across different concentrations (0.1, 1, and 10 µg/mL), there was no consistent pattern of decreasing nitrite levels with increasing concentrations. Crude extract at 1 µg/mL caused the highest nitrite level, while other concentrations did not follow a clear trend (Figure 4c). This lack of dose dependency suggests that the bioactive compounds within the crude extracts may interact in a complex manner, potentially involving synergistic or antagonistic effects that are not strictly linear with concentration. Therefore, while the crude extracts demonstrated anti-inflammatory properties, the absence of a dose-dependent response highlights the complexity of the phytochemical interactions and shows the need for further investigation to fully understand the bioactivity mechanisms.

Our results suggest that both the crude extract and gomphrenin enriched fraction from B. alba have the capacity to reduce nitric oxide production, compared to LPS and standard antioxidants such as meloxicam and indomethacin. The gomphrenin enriched fraction, in particular, showed a greater ability to decrease nitrite levels compared to the crude extract, underscoring their potential as effective anti-inflammatory agents. The Griess test results support the anti-inflammatory potential of the gomphrenin enriched fraction in a concentration-dependent manner, unlike the crude extract of *B. alba* f. *rubra*.

For further elucidation of the anti-inflammatory effects of the crude extract and gomphrenin enriched fraction of *B. alba* f. *rubra*, we measured cyclooxygenase (COX) activity. The COX enzymes, particularly COX-2, are crucial in the inflammatory response as they catalyze the conversion of arachidonic acid to prostaglandins, which mediate inflammation and pain [20].

The COX activity was evaluated using a COX inhibitor screening assay. This assay quantifies the production of prostaglandin E2 (PGE2) in cells treated with the extracts and LPS. The assay involved incubating cells with arachidonic acid and measured the PGE2 production using an enzyme-linked immunosorbent assay (ELISA). The level of PGE2 was directly proportional to COX activity, allowing us to assess the inhibitory effects of the treatments on COX enzymes. The effects of the crude extract (B) and gomphrenin enriched fraction (A) on COX activity were assessed relative to the LPS-induced control group, which showed a marked increase in COX activity, confirming the induction of an inflammatory response.

The enriched gomphrenin fraction showed concentration-dependent activity in reducing the COX activity, while the crude extract showed the opposite results (Figure 4d).

Betalains have been found to inhibit inflammatory mediators like cyclooxygenase-2 (COX-2) and lipoxygenase, which are crucial in the biosynthesis of pro-inflammatory prostaglandins and leukotrienes. Previous studies have shown that betanin from beets (Amaranthaceae) and opuntia (Cactaceae) were able to reduce COX-2 activity, thereby lowering inflammation [21]. Our research also indicates such possibilities of both the *B. alba* f. *rubra* crude extract and the gomphrenin enriched fraction.

The analysis of cytokine secretion revealed insightful observations regarding the anti-inflammatory properties of the crude extract and gomphrenin enriched fraction of *B. alba* f. *rubra*. The results were compared to the LPS control, which serves as a positive control for inflammation induction. The LPS treatment significantly increased IL-1β secretion, with the levels peaking at approximately 1.0 pg/mL for the 1 h treatment and remaining even higher for the 24 h treatment. In comparison, all other treatments, including the crude extract and enriched fraction, exhibited markedly lower IL-1β levels. Despite the various concentrations tested (0.1 to 10 µg/mL), there was no evident dose-dependent effect of all tested compounds, including reference substances, on IL-1β secretion (Figure 5a). For instance, at 0.1 µg/mL, both gomphrenin fraction A, crude extract B, and Mel showed relatively low IL-1β levels, but higher concentrations did not consistently inhibit the secretion of IL-1β.

In case of IL-6, the cytokine levels in both A- and B-treated cells as well as in reference substances treatments were lower than the control medium and significantly decreased compared to LPS-stimulated cells, although again no clear concentration-dependent effect was observed (Figure 5b). For instance, at 0.1 µg/mL, IL-6 levels were relatively low, but increasing the concentration up to 10 µg/mL did not consistently lower IL-6 levels.

The cytokine secretion assays indicated the moderate anti-inflammatory potential of both the crude extract (B) and gomphrenin enriched fraction (A) compared to all reference substances routinely used as anti-inflammatory drugs, as well as compared to the robust inflammatory response induced by LPS. The higher IL-1β secretion in the response to LPS treatment may suggest a specific inflammatory response pathway activation, as IL-1β is a key mediator of the inflammatory response. The fact that the crude extract and gomphrenin enriched fraction could decrease IL-1β levels indicate their potential anti-inflammatory effects. The lack of concentration dependence prompts further research into the mechanism of action of compounds with anti-inflammatory potential in this cellular model, especially since these observations also apply to reference substances.

To evaluate the anti-inflammatory effects of the crude extract and gomphrenin enriched fraction, we assessed the activity of nuclear factor-kappa B (NF-κB), a key transcription factor involved in inflammatory responses. The activation of NF-κB was measured using an NF-κB assay. The assay involves cells transfected with a construct containing NF-κB binding sites linked to a luciferase gene. Upon activation, NF-κB translocates to the nucleus, binds to these sites, and drives the expression of the luciferase gene. The level of luciferase activity, quantified using a luminometer, is directly proportional to the activity of NF-κB.

In our study, the effects of crude extract (B) and a gomphrenin enriched fraction (A) on NF-κB activation were evaluated in the context of LPS-induced inflammatory response (Figure 5c). The control group treated with LPS showed a significant increase in NF-κB activity, confirming the successful induction of an inflammatory response.

At lower concentrations (0.1 µg/mL and 1 µg/mL), the crude extract showed a modest reduction in NF-κB activity compared to the LPS control, while its higher concentration (10 µg/mL) demonstrated a more noticeable decrease in NF-κB activation, suggesting a potential dose-dependent inhibition. The gomphrenin enriched fraction (A) at all concentrations exhibited a slight decrease in NF-κB activity, with no concentration-dependent response.

To achieve a comprehensive understanding of the anti-inflammatory potential of gomphrenins from *B. alba* f. *rubra*, we employed a variety of analytical and biological methods. Phytochemical characterization was performed using LC-DAD-ESI-MS/MS to identify and quantify the gomphrenin pigments in plant extracts [1]. The cytotoxicity and anti-inflammatory effect of the gomphrenin enriched fraction and crude extract were tested using human and animal cell models. We utilized normal human dermal fibroblasts (NHDFs), mouse fibroblasts (L929), and differentiated human acute leukemia monocyte (THP-1) cells. Various assays, including SRB, DCF-DA, Griess, MDA, and ELISA, were used to measure cell viability, oxidative stress markers, and cytokine levels, offering comprehensive insights into the bioactivity of gomphrenins.

Considering the results obtained in our study, the strong antioxidant properties of betalains, both as an enriched fraction and contained in the crude extract, helped to neutralize reactive oxygen species (ROS) and reduce oxidative stress and subsequent inflammation (Figure 1, Figure 2, Figure 3, Figure 4 and Figure 5). Betalains possess the ability to scavenge free radicals and chelate metal ions, thereby preventing oxidative stress and cellular damage, as was previously shown for Opuntia ficus-indica plants [22,23]. The antioxidant activity of the enriched gomphrenin fraction as well as crude B. alba fruit extract was previously noted, only using the ABTS, FRAP, and ORAC assays, in relation to the activity of caffeic acid [1]. A significantly higher antioxidant activity of betalains was also observed compared to other natural pigments like anthocyanins [24].

Data obtained in the DCF-DA assay, in our study, revealed that the crude extract had a superior ability to reduce ROS levels compared to the gomphrenin enriched fraction. This highlights its potential as an antioxidant, although linked to the plant matrix being rich not only in gompherins but also in other specialized metabolites. Thus, it can be speculated that other compounds besides gompherins are involved in mitigating oxidative stress, such as polyphenolic compounds, especially due to the fact that A and B contained both groups of compounds. It was previously found that polyphenolic compounds contained in water extracts are well-known free radical scavengers [25].

Similarly, the MDA assay results supported the higher antioxidative potential of the extract compared to the gomphrenin enriched fraction of *B. alba* f. *rubra*. The results for the crude extract demonstrated its significant ability to reduce lipid peroxidation, showcasing its promise in protecting against oxidative stress and associated inflammation. Our results are also consistent with those obtained so far for this group of compounds obtained from other plant species such as *Amaranthus tricolor* L. (Amaranthaceae) [26]. In the study on *A. tricolor*, a betalain-rich extract demonstrated significant antioxidant and anti-inflammatory effects by reducing oxidative damage [27].

Moreover, our research showed that in the case of tests for NF-κB inhibitory activity, both the gomphrenin enriched fraction and the crude extract from *B. alba* f. *rubra* fruits were more effective than standardly used non-steroidal anti-inflammatory drugs, such as indomethacin and maloxicam, but they worked less effectively than pure ascorbic acid (Figure 5b). Betalains have been found to inhibit the activation of NF-κB, leading to a decrease in the secretion of various cytokines such as interleukin-6 (IL-6) and IL-1*β* in inflammatory models, i.e., macrophages, which are crucial cells in the immune response. This effect has been observed in various plant extracts, including those from the Cactaceae (*Opuntia ficus-indica*), Amaranthaceae (*Gomphrena globosa*) and Nyctaginaceae (*Mirabilis jalapa*) families [23,28,29]. Although betalains are primarily antioxidants, they can exhibit pro-oxidant activity in high concentrations or specific environments [6]. As mentioned, the effects of betalains can vary depending on the cellular context and concentration. Our studies also confirmed that moderate concentrations exerted anti-inflammatory effects, while higher concentrations activated inflammatory pathways, leading to pro-inflammatory outcomes.

By targeting multiple pathways, gomphrenins exhibit a multifaceted mode of action, making them promising candidates for therapeutic interventions in conditions driven by inflammation and oxidative stress. The biological activity of these natural compounds can therefore be expressed by possible mechanisms of action such as the inhibition of pro-inflammatory enzymes like COX1 and COX2 and, potentially, LOX; the suppression of key inflammatory cytokines (e.g., IL-6, IL-1β); a reduction in nitric oxide (NO) production by downregulating iNOS; scavenging reactive oxygen species (ROS) to lower oxidative stress; decreasing lipid peroxidation and protecting cellular membranes; NF-κB pathway modulation by inhibition of NF-κB activation; and a reduction in the expression of inflammation-related genes.

Future research should focus on in vivo studies using animal and human models, which are needed to understand the bioavailability, pharmacokinetics, and systemic effects of betalains. However, due to the complexity of physiological environments and interactions in living organisms, our findings may not fully translate to in vivo systems. The use of phytochemically characterized crude extracts and enriched gomphrenin fractions may introduce variability in results due to the presence of other bioactive compounds, making it difficult to isolate the effects of gomphrenins alone. Although the dual role of betalains is acknowledged, future studies could benefit from a more extensive investigation into the threshold at which gomphrenins shift from exerting anti-inflammatory to pro-inflammatory effects. Also, this study emphasizes the acute effects of gomphrenins on inflammation and antioxidative stress but does not address their long-term impact or potential toxicity with prolonged exposure. Addressing these limitations in the future, along with a detailed exploration of the molecular pathways influenced by gomphrenins, will help delineate their anti- and pro-inflammatory actions and could provide a more comprehensive understanding of these substances’ therapeutic potential and safety profile.

## 4. Conclusions

In summary, betalains, particularly gomphrenins, emerge as promising candidates for therapeutic applications, offering mechanisms such as enzyme inhibition, antioxidant effects, and inflammatory pathway modulation. The comprehensive bioactivity study of a gomphrenin enriched extract from *B. alba* f. *rubra* using various analytical and biological methods highlighted its significant anti-inflammatory and antioxidative properties. Both the crude extract and gomphrenin enriched fraction demonstrated the ability to inhibit pro-inflammatory cytokines, reduce oxidative stress markers, and mitigate lipid peroxidation. The gomphrenin enriched fractions, in particular, showed superior efficacy in reducing ROS and NO levels, underscoring their potential as effective anti-inflammatory and antioxidant agents. This dual functionality can be particularly beneficial in the context of chronic inflammatory diseases, where oxidative stress often plays a critical role in disease progression. The findings also support the therapeutic potential of *B*. *alba* f. *rubra* extracts and warrant further investigation into their mechanisms of action and broader applications in inflammation-related conditions. Betalains, and, in particular, betacyanins such as gomphrenins, produced in Basellaceae, Cactaceae, Nyctaginaceae, and Amaranthaceae plants, exhibit significant bioactive potential, encompassing antioxidant and anti-inflammatory properties. Their diverse mechanisms of action and therapeutic promise highlight the need for further research to fully harness their benefits in clinical settings. The anti-inflammatory effects of gomphrenins and other betalains offer significant promise in the management of inflammation-related diseases. Their multifaceted mechanisms, including enzyme inhibition, antioxidant activity, and modulation of inflammatory pathways, underpin their therapeutic potential.

## Figures and Tables

**Figure 1 nutrients-16-04393-f001:**
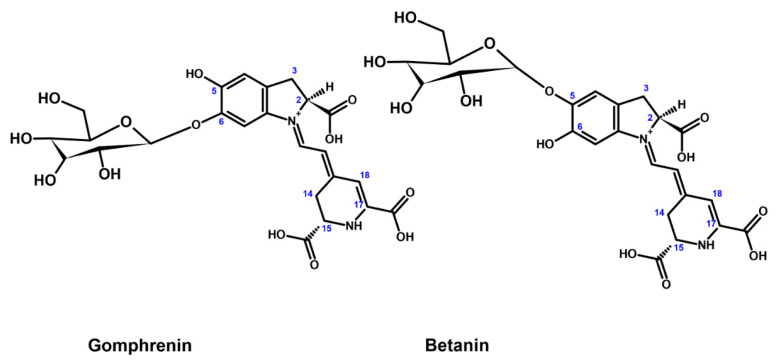
Chemical structures of gomphrenin (glycosylated at carbon 6) and betanin (more commonly found glycosylation at carbon 5).

**Figure 2 nutrients-16-04393-f002:**
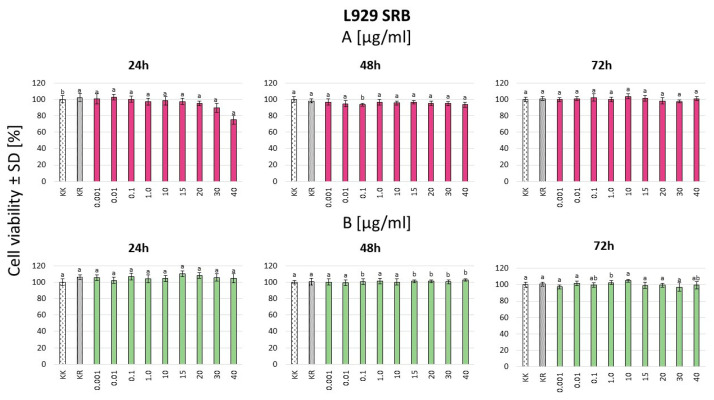
L929 cell line SRB test for gomphrenin enriched fraction (**A**) and crude extract (**B**). KK—cell culture in a culture medium; KR—culture fluid without the presence of cells. Statistical significance of differences was estimated using the Kruskal–Wallis test at *p* < 0.05 and is indicated on the figure—the same letters indicate lack of statistical significance of differences.

**Figure 3 nutrients-16-04393-f003:**
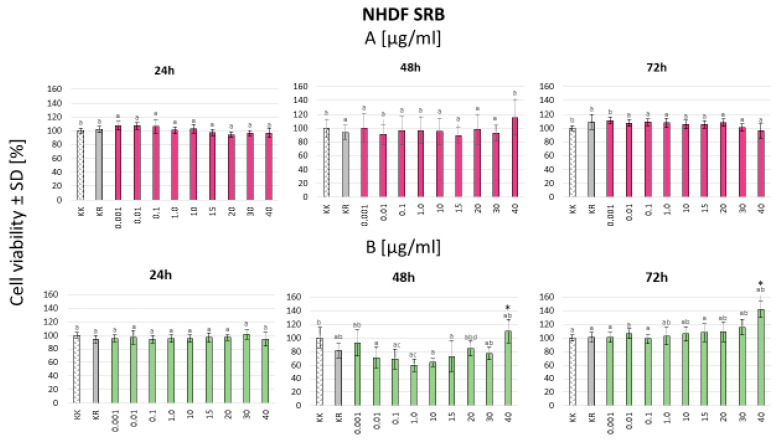
NHDF cell line SRB test for gomphrenin enriched fraction (**A**) and crude extract (**B**). KK—cell culture in a culture medium; KR—culture fluid without the presence of cells. Statistical significance of differences was estimated using the Kruskal–Wallis test at *p* < 0.05 (*) (the same lowercase letters above the bars indicate lack of significant differences).

**Figure 4 nutrients-16-04393-f004:**
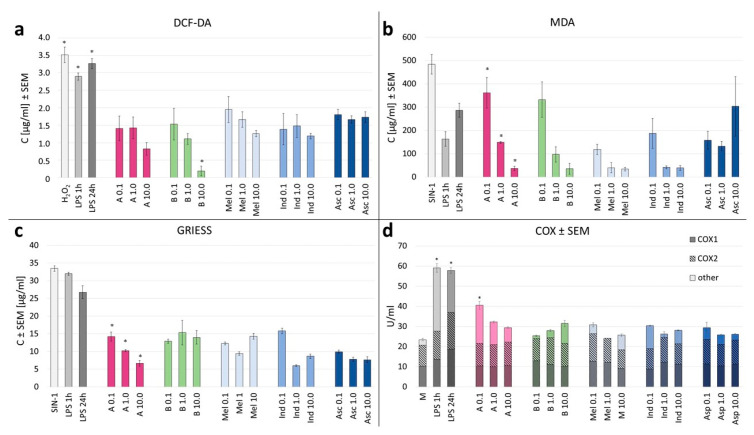
(**a**–**d**) The effect of betalain enriched fraction (A), crude extract (B), meloxicam (Mel), indomethacin (Ind), ascorbic acid (Asc), aspirin (Asp), and lipopolysaccharide (LPS) on ROS secretion in THP-1 cell model using DCF-DA assay (**a**); MDA assay (**b**); Griess assay (**c**); COX activity assay (**d**) with different bar shading indicating COX-1 and COX -2 isoforms and non-specific activity. Statistical significance of differences was estimated using the Kruskal–Wallis test at *p* < 0.05 (*).

**Figure 5 nutrients-16-04393-f005:**
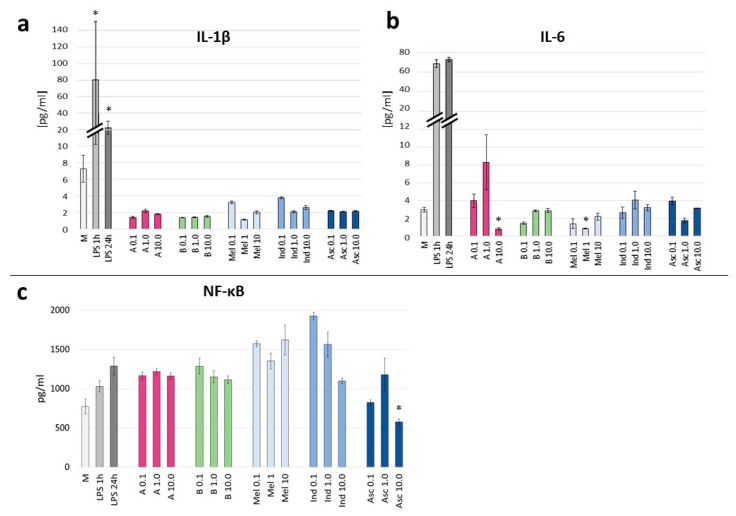
(**a**–**c**) The effect of enriched gomphrenin fraction (A) and crude extract on cytokines IL-1β (**a**) and IL-6 (**b**) and transcription factor NF-κB (**c**) secretion in LPS-stimulated THP-1 cell model. Meloxicam (Mel), indomethacin (Ind), ascorbic acid (Asc), medium (M). Statistical significance of differences was estimated using the Kruskal–Wallis test at *p* < 0.05 (*).

## Data Availability

All data obtained in this study are contained in the article and the Appendix A as well as available upon request from the authors.

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
