# Peer review of "Anti-Inflammatory Activity of Gomphrenin-Rich Fraction from Basella alba L. f. rubra Fruits"

_nutrients, 2024, doi:10.3390/nu16244393_

Round 1
Reviewer 1 Report
Comments and Suggestions for Authors
This manuscript is about the anti-inflammatory and antioxidant benefits of Basella alba L. f. ‘rubra’, focusing on its fruit extracts, which are rich in natural compounds called gomphrenins. The authors tested both a crude extract and a purified gomphrenin fraction using various lab methods. Both extracts reduced inflammation and oxidative stress by lowering harmful molecules like ROS, NO, and pro-inflammatory cytokines. The purified gomphrenin fraction showed stronger effects, especially in reducing oxidative stress and inflammation markers. These findings suggest that B. alba fruit extracts, especially the enriched fraction, have potential as natural treatments for inflammation-related conditions. Further research is needed to understand how they work and explore their uses.
It is clear that a lot of experimental work has been done and the authors have put in a lot of effort. The article is interesting and will definitely be of interest to the guild in this area.
Anyway, I have some critical suggestions and questions:
1. The abstract is too long, please shorten it.
2. Please correct the anti-oxidant to antioxidant. Both "antioxidant" and "anti-oxidant" are correct, but "antioxidant" is the more commonly used and widely accepted spelling in scientific and general usage.
3. Row 84, remove the dot before the reference.
4. The manuscript does not specify the number of replicates for each experiment- sample size and replication. Including this information is crucial for assessing statistical validity.
5. While various controls are mentioned, there is a lack of clarity regarding how these group controls were selected and their relevance to the experimental design.
6. The procedure for protective properties against ROS or NO formation is not clearly presented. Please add a detailed explanation.
7. You tested a wide range of concentrations for the gomphrenin fraction and crude extract. What criteria did you use to determine the specific concentration ranges (0.0001 - 40 μg/ml) for your assays?
8. Can you provide more details on the phytochemical characterization of the gomphrenin-rich fraction? What specific methods were used to confirm the presence and concentration of gomphrenins in your extracts?
9. Please format the references according to the journal's required style.
Based on the above I recommend a major revision.
Comments on the Quality of English LanguageThe English language is fine, minor revision is required.
Author Response
Dear Reviewer, We appreciate all the effort paid to evaluate our manuscript and all the useful and constructive comments. Below, we have responded to each of the comments and explained how the manuscript has been revised. We do hope, that in the present form, the submission meets the Reviewer's expectations.
Comment 1. The abstract is too long, please shorten it.
Response 1. Thank you for this suggestion. The abstract has been shortened.
Comment 2. Please correct the anti-oxidant to antioxidant. Both "antioxidant" and "anti-oxidant" are correct, but "antioxidant" is the more commonly used and widely accepted spelling in scientific and general usage.
Response 2. Thank you for this comment. The expression ‘anti-oxidant’ has been replaced by ‘antioxidant’ in the manuscript.
Comment 3. Row 84, remove the dot before the reference.
Response 3. Thank you. The dot was removed.
Comment 4. The manuscript does not specify the number of replicates for each experiment- sample size and replication. Including this information is crucial for assessing statistical validity.
Response 4. Thank you for pointing this out. We have clarified that all experiments were performed in four independent biological replicates, each with four technical replicates to ensure statistical validity. For ELISA assays, the material was prepared from four independent replicates, with duplicate or triplicate measurements on ELISA plates as per the manufacturer’s instructions. This information has been added to the "Statistical Analysis" section.
Comment 5. While various controls are mentioned, there is a lack of clarity regarding how these group controls were selected and their relevance to the experimental design.
Response 5. Dear Reviewer, we appreciate this observation. Additional details have been included in the "Materials and Methods" section to explain the selection and relevance of controls. Specifically, lipopolysaccharide (LPS) was used as a positive control to induce inflammation, while meloxicam, indomethacin, and ascorbic acid served as reference anti-inflammatory and antioxidant agents. Blank tests (KK) and culture fluid without cells (KR) were used to assess baseline activity and ensure no nonspecific effects. These controls were chosen to reflect conditions relevant to the experimental design and to allow for the benchmarking of the tested compounds' bioactivity. We hope you find this explanation satisfactory.
Comment 6. The procedure for protective properties against ROS or NO formation is not clearly presented. Please add a detailed explanation.
Response 6. Thank you for this suggestion. The procedure for assessing protective properties against ROS and NO formation has been expanded in the "Materials and Methods" section. Briefly, cells were pre-incubated with test compounds before oxidative or nitrosative stress induction using Hâ‚‚Oâ‚‚ (100 μM) or SIN-1 (100 μM), respectively. ROS levels were assessed using the DCF-DA assay, and NO levels were measured using the Griess assay. These procedures were designed to evaluate the ability of the tested compounds to mitigate oxidative and nitrosative stress effectively. It is our sincere hope that this explanation will meet with your approval.
Comment 7. You tested a wide range of concentrations for the gomphrenin fraction and crude extract. What criteria did you use to determine the specific concentration ranges (0.0001 - 40 μg/ml) for your assays?
Response 7. Thank you for this question. The selected concentration range was based on preliminary cytotoxicity assays, which identified non-toxic levels, and on data from the literature, ensuring the inclusion of biologically relevant and sub-toxic concentrations. The lower concentrations (0.0001–0.01 μg/ml) were included to investigate potential bioactivity at trace levels, while higher concentrations (up to 40 μg/ml) enabled evaluation of maximum efficacy and dose-response relationships. This rationale has been incorporated into the "Materials and Methods" section. We hope you find this explanation satisfactory.
Comment 8. Can you provide more details on the phytochemical characterization of the gomphrenin-rich fraction? What specific methods were used to confirm the presence and concentration of gomphrenins in your extracts?
Response 8. More details are added to section: 2.1. Plant material, substances and reagents used in assays:
The samples were submitted to spectrophotometric as well as LC-MS analyses [1]. For the measurement of the total concentration of the pigments, the extracts were analyzed spectrophotometrically. The total concentration was expressed as mg gomhrenin equivalents/100 g of fresh fruits. Quantification of betacyanins was evaluated taking a molar extinction coefficient of ε = 65,000 M−1 cm−1 at 539 nm for betanin in spectrophotometric calculations [new ref]. The betacyanin profiles were established by LC-MS analyses according to previous report [1].
[new ref] Schwartz, S.J.; von Elbe, J.H. Quantitative determination of individual betacyanin pigments by high-performance liquid chromatography. J. Agric. Food Chem. 1980, 28, 540–543.
Comment 9. Please format the references according to the journal's required style.
Response 9. Thank you. The references were formatted according to the journal's required style.
All new paragraphs are visible in 'track changes', and clarifications to reviewers' comments in the text are highlighted in yellow.
Reviewer 2 Report
Comments and Suggestions for Authors
Areas for Improvement:
- Clarity in abstract: The abstract could be more concise and focused. Consider rephrasing to emphasize the most important findings and their implications.
- Specificity of effects: While the study shows anti-inflammatory effects, it also notes that the effects are not always consistent or dose-dependent. This should be addressed more explicitly in the discussion, exploring potential reasons for this variability.
- Mechanistic insights: The study primarily focuses on observing the effects of the extracts. Further investigation into the underlying mechanisms of action would strengthen the manuscript.
- Clinical relevance: While the in vitro results are promising, the authors should discuss the limitations of the study and the need for further research, including in vivo studies, to assess the clinical relevance of these findings.
- Minor textual inconsistencies: There are some minor inconsistencies in the text (e.g., "Figur 6" instead of "Figure 6"). A careful proofread is recommended.
Specific Comments:
- Introduction: Clearly define the research gap and how this study addresses it.
- Materials and Methods:
- Provide more details on the plant material, including cultivation conditions and fruit collection methods.
- Clarify the rationale for using different cell lines.
- Ensure all reagents and equipment are properly cited.
- Results:
- Consider presenting the results in a more concise manner, potentially combining some figures.
- Provide more detailed statistical analysis, including effect sizes and confidence intervals.
- Discussion:
- Expand the discussion on the potential mechanisms of action of gomphrenins.
- Address the limitations of the study and suggest future research directions.
- Conclusion: Summarize the key findings and their implications for the field.
Author Response
Comments and Suggestions for Authors
Areas for Improvement:
Comment 1. Clarity in abstract: The abstract could be more concise and focused. Consider rephrasing to emphasize the most important findings and their implications.
Response 1. We have modified the abstract for conciseness. We hope that it now conveys a clearer message
Comment 2. Specificity of effects: While the study shows anti-inflammatory effects, it also notes that the effects are not always consistent or dose-dependent. This should be addressed more explicitly in the discussion, exploring potential reasons for this variability.
Response 2. Yes, indeed the results of some assays were quite variable and without obvious trends. In our opinion, there may be various mechanisms involved, but one of the major reason would be the influence of the extract matrix and a presence of various compounds that could act in different directions, causing some degree of instability and unpredictability. This kind of effect is known in studies involving extracts or purified fractions. his issue has been addressed in the revised discussion.
Comment 3. Mechanistic insights: The study primarily focuses on observing the effects of the extracts. Further investigation into the underlying mechanisms of action would strengthen the manuscript.
Response 3. We totally agree that more mechanistic insights would add a lot to the study. However, we have to limit the study to the submitted results. To extend the studies to the mechanisms, we are in the process of isolating larger amounts of gomphrenins to reduce the matrix effects and allow to dissect the specific mechanisms (like cell signaling or enzymological interactions). For such studies, enough compounds is needed and betacyanins are a tricky class of compounds to obtain. Furthermore, the in silico studies, such as molecular docking could prove useful but only in concert with verified experimental results. Nevertheless, some mechanistic considerations were included in the discussion.
Comment 4. Clinical relevance: While the in vitro results are promising, the authors should discuss the limitations of the study and the need for further research, including in vivo studies, to assess the clinical relevance of these findings.
Response 4. Indeed, the clinical relevance of this approach is rather limited without translation to in vivo models. However, as there are still gaps in understanding of the observed mechanisms, the priority would rather be on the deeper understanding before forwarding the compounds to in vivo on more advanced animal or human studies. Alternatively, some less complex animal models (like C. elegans) are envisaged but it requires a separate project. Nonetheless, we appreciate this suggestion. finally, the limitations of the current study have been addressed in the revised manuscript.
Comment 5. Minor textual inconsistencies: There are some minor inconsistencies in the text (e.g., "Figur 6" instead of "Figure 6"). A careful proofread is recommended.
Response 5. The revised manuscript has been checked. In addition, the Figures 4-9 were combined. The results for the assays: DCF-DA, MDA, GRIESS and COX were shown in Figure 4 with the corresponding letters 'a' to 'd', and for the cytokines and NFκB were shown on Figure 5 with the corresponding letters 'a' to ‘c’.
Specific Comments:
Comment 6. Introduction: Clearly define the research gap and how this study addresses it.
Response 6. The Introduction section was improved by adding sentences explaining the purpose and motive for undertaking the research presented in the study.
‘The existing studies predominantly focus on general betalain types or specific sources, such as beet-derived betanin, with limited exploration of less common beta-lain-rich sources. Thus, in this study, we have investigated the anti-inflammatory properties of phytochemically characterized crude extract and enriched fraction of gomphrenins obtained from Basella alba f. ‘rubra’, a red-stem morphological form, known for its high betalain content. The cytotoxicity and anti-inflammatory effects of the extracts and gomphrenins were assessed using normal human dermal fibroblasts (NHDF), mouse fibroblasts (L929), and differentiated human acute leukemia monocyte (THP-1) cells. Various assays, including SRB, DCF-DA, Griess, MDA, and ELISA, were employed to evaluate cell viability, oxidative stress markers, and cytokine levels, providing a detailed insight into the bioactivity of gomphrenins. A dual role of beta-lains, were higher concentrations may induce pro-inflammatory effects, remains un-derexplored, especially in the context of therapeutic applications where dose optimi-zation is crucial.’
Comment 7. Materials and Methods: Provide more details on the plant material, including cultivation conditions and fruit collection methods.
Response 7. More details on the plant material were added to section: 2.1. Plant material, substances and reagents used in assays:
“Sowing of the seeds was performed in a 3:1 ratio of soil and coconut pith mass and watered daily. The seedlings were transplanted to fertile soil with plenty of organic matter and a pH of 6.5–6.8. The plants were designed to support the fast climbing of the vines which were trellised so that they reached up to 3 m in length at controlled humidity and temperature, ensuring continuous flowering and fruiting [1].”
Comment 8. Clarify the rationale for using different cell lines.
Response 8. Thank you for this comment. The use of different cell lines in this study was strategically planned to ensure a comprehensive evaluation of the biological effects of the gomphrenin-enriched fraction and crude extract. Each cell line was chosen for its specific relevance to the experimental goals:
- Normal Human Dermal Fibroblasts (NHDF):
NHDF cells represent a model of normal human cells. This allows for the evaluation of the potential cytotoxic effects of the tested compounds under conditions mimicking normal physiological environments. To ensure the tested concentrations were non-toxic to healthy human cells.
- Mouse Fibroblasts (L929):
L929 cells are robust and widely used in cytotoxicity and recommended by norm ISO 10993 part V. To provide a comparative model for cytotoxicity complementing the data from NHDF cells.
- Differentiated THP-1 Cells (Human Acute Leukemia Monocytes):
Differentiated THP-1 cells serve as a macrophage-like model and are widely recognized for their utility in studying inflammatory responses. These cells can be stimulated with lipopolysaccharide (LPS) to mimic inflammation and oxidative stress, providing a relevant system for testing anti-inflammatory agents. To investigate the modulation of key inflammatory markers (e.g., IL-1β, IL-6, NF-κB) and oxidative stress parameters (ROS, NO) by the tested compounds in an inflammation-specific context.
The use of these three cell lines allowed for a multi-dimensional assessment of the tested compounds: NHDF and L929 provided insights into non-toxic, offered a robust system for cytotoxicity; and THP-1 cells served as a model for inflammation-specific bioactivity. This combination ensured that the study could evaluate the safety, antioxidant potential, and anti-inflammatory properties of the gomphrenin fraction and crude extract in a contextually relevant manner.
We hope you find this explanation satisfactory.
Comment 9. Ensure all reagents and equipment are properly cited.
Response 9. Thank you for the suggestion. We have carefully reviewed the manuscript and ensured that all reagents and equipment are properly cited, including manufacturer details and product specifications, where applicable. These updates have been incorporated into the "Materials and Methods" section to meet the required standards.
Comment 10. Results: Consider presenting the results in a more concise manner, potentially combining some figures. Provide more detailed statistical analysis, including effect sizes and confidence intervals.
Response 10. Thank you for this suggestion. We have reviewed the manuscript and added the necessary information.
The effect sizes were calculated to show the strength of the relationship between tested compounds and control/reference compound. The confidence intervals (CI) were calculated and the results are given in the Table 1S in the supplementary files.
Figures were combined. The results for the assays: DCF-DA, MDA, GRIESS and COX were shown in Figure 4 with the corresponding letters 'a' to 'd', and for the cytokines and NFκB were shown on Figure 5 with the corresponding letters 'a' to ‘c’.
Additionally, the results of the effect of the tested compounds on cyclooxygenase activity were presented graphically in Figure 4d in a corrected form, i.e. separately for COX 1 and COX 2 and other possible cyclooxygenase isoforms. An appropriate description is provided in the Materials and Methods section, subsection 2.13 COX activity.
Comment 11. Discussion: Expand the discussion on the potential mechanisms of action of gomphrenins. Address the limitations of the study and suggest future research directions.
Response 11. Thank you for this suggestion. The potential mechanisms of action of gomphrenins, a s well as the limitations of the study and future research directions have been included in the Discussion section:
“By targeting multiple pathways, gomphrenins exhibit a multifaceted mode of ac-tion, making them promising candidates for therapeutic interventions in conditions driven by inflammation and oxidative stress. The biological activity of these natural compounds can therefore be expressed by such possible mechanisms of action as: inhi-bition of pro-inflammatory enzymes like COX1 and COX2, and potentially LOX; sup-pression of key inflammatory cytokines (e.g., IL-6, IL-1β); reduction of nitric oxide (NO) production by downregulating iNOS; scavenging reactive oxygen species (ROS) to lower oxidative stress; lipid peroxidation decrease and cellular membranes pro-tection; NF-κB pathway modulation by inhibition of NF-κB activation, and the reduc-tion of inflammation-related genes expression.
Future research should focus on in vivo studies using animal and human stud-iessystems, which are needed to understand the bioavailability, pharmacokinetics, and systemic effects of betalains. However, due to the complexity of physiological environments and interactions in living organisms, our findings may not fully translate to in vivo systems. The use of phytochemically characterized crude extracts and enriched gomphrenin fractions may introduce variability in results due to the presence of other bioactive compounds, making it difficult to isolate the effects of gomphrenins alone. Although the dual role of betalains is acknowledged, the future study could benefit from more extensive investigation into the threshold at which gomphrenins shift from anti-inflammatory to pro-inflammatory effects. Identifying the optimal dosage that maximizes anti-inflammatory effects while minimizing any pro-inflammatory risks. Also, the study emphasizes acute effects of gomphrenins on inflammation and antioxidative stress but does not address their long-term impact or potential toxicity with prolonged exposure. Addressing these limitations in the future along with detailed exploration of the molecular pathways influenced by gomphrenins will help delineate their anti- and pro-inflammatory actions and could provide a more comprehensive understanding of these substances’ therapeutic potential and safety profile.”
Comment 12. Conclusion: Summarize the key findings and their implications for the field.
Response 12 . Thank you for this suggestion. The summarized key findings and their implications for the field have been included in the Conclusion section:
“In summary, betalains, particularly gomphrenins, emerge as promising candidates for therapeutic applications, offering mechanisms such as enzyme inhibition, antioxidant effects, and inflammatory pathway modulation. Tthe comprehensive bioactivity study of gomphrenin enriched extract from B. alba f. ‘rubra’ using various analytical and biological methods highlighted their significant anti-inflammatory and antioxidative properties. Both the crude extract and gom-phrenin enriched fraction demonstrated the ability to inhibit pro-inflammatory cyto-kines, reduce oxidative stress markers, and mitigate lipid peroxidation. The gomphrenin enriched fractions showed superior efficacy in reducing ROS and NO levels, underscoring their potential as effective anti-inflammatory and anti-oxidant agents. This dual functionality can be particularly beneficial in the context of chronic inflammatory diseases, where oxidative stress often plays a critical role in dis-ease progression. The findings also support the therapeutic potential of B. alba f. ‘rubra’ extracts and warrant further investigation into their mechanisms of action and broad-er applications in inflammation-related conditions. Betalains, and in particular betacyanins such as gomphrenins, produced in Basellaceae, Cactaceae, Nyctaginaceae, and Amaranthaceae plants, exhibit significant bioactive potential, encompassing antioxi-dant, and anti-inflammatory properties. Their diverse mechanisms of action and therapeutic promise highlight the need for further research to fully harness their bene-fits in clinical settings. The anti-inflammatory effects of gomphrenins and other betalains offer significant promise in the management of inflammation-related diseases. Their multifaceted mechanisms, including enzyme inhibition, antioxidant activity, and modulation of inflammatory pathways, underpin their therapeutic potential.”
All new paragraphs are visible in 'track changes', and clarifications to reviewers' comments in the text are highlighted in yellow.
Finally, we would kindly like to acknowledge the Reviewer's effort paid to evaluation of our submission and the constructive comments that were very useful for improvement of the manuscript.
Round 2
Reviewer 1 Report
Comments and Suggestions for Authors
Now the manuscript looks much better.